# Isolation of Small Extracellular Vesicles from Human Sera

**DOI:** 10.3390/ijms22094653

**Published:** 2021-04-28

**Authors:** Małgorzata S. Małys, Christof Aigner, Stefan M. Schulz, Helga Schachner, Andrew J. Rees, Renate Kain

**Affiliations:** 1Department of Pathology, Medical University Vienna, 1090 Vienna, Austria; malgorzata.malys@meduniwien.ac.at (M.S.M.); christof.aigner@meduniwien.ac.at (C.A.); stefan.schulz@meduniwien.ac.at (S.M.S.); helga.schachner@meduniwien.ac.at (H.S.); andrew.rees@meduniwien.ac.at (A.J.R.); 2Division of Nephrology and Dialysis, Department of Medicine III, Medical University Vienna, 1090 Vienna, Austria

**Keywords:** extracellular vesicles, exosomes, purification, isolation

## Abstract

Robust, well-characterized methods for purifying small extracellular vesicles (sEV) from blood are needed before their potential as disease biomarkers can be realized. Here, we compared isolation of sEV from serum by differential ultracentrifugation (DUC) and by exclusion chromatography using commercially available Exo-spin™ columns. We show that sEV can be purified by both methods but Exo-spin™ columns contain copious additional particles recorded by nanoparticle tracking analysis, invalidating its use for quantifying yields. DUC samples contained higher concentrations of exosome specific proteins CD9, CD63 and CD81 and electron microscopy confirmed that most particles in DUC preparations were sEV, whereas Exo-spin™ samples also contained copious co-purified plasma lipids. MACSPlex bead analysis identified multiple exosome surface proteins, with stronger signals in DUC samples, enabling detection of 21 of 37, compared to only 10 in Exo-spin™ samples. Nevertheless, the pattern of expression was consistent in both preparations, indicating that lipids do not interfere with bead-based technologies. Thus, both DUC and Exo-spin™ can be used to isolate sEV from human serum and what is most appropriate depends on the subsequent use of sEV. In summary, Exo-spin™ enables isolation of sEV from blood with vesicle populations similar to the ones recovered by DUC, but with lower concentrations.

## 1. Introduction

Extracellular vesicles (EV) are a heterogeneous group of a lipid bound structures released from cells into the surrounding medium [1]. The number and composition of EV depends on the stress and activation status of the originating cell, and this provides the basis for their critical role in cell–cell communication during homeostasis and after injury. They have often been isolated from tissue culture supernatants and less commonly from the blood of donors, including those with systemic disease, in whom they are especially abundant. This raises the possibility of using blood derived EV to probe the pathogenesis of disease, as biomarkers of its activity, and even as novel vehicles to deliver therapy.

EV can be classified into three distinct types, defined by their biogenesis [1]: apoptopic bodies whose diameters range between 100–5000 nm, and which originate as blebs that detach from the surface of apoptotic cells; microvesicles, with diameters of 100–1000 nm, which are formed by budding out from the plasma membrane; and exosomes, of 30–150 nm, which are generated as intraluminal vesicles in late endosomes that fuse with the plasma membrane to release the ILV into the extracellular fluid as exosomes [2]. Despite their differing origins, the three species of EV have overlapping physical properties and share some of the molecules expressed on their limiting membranes, and also components of the cargo they carry. These overlapping properties have hampered the development of strategies to obtain pure preparations of the individual types of EV, and for this reason the International Society for Extracellular Vesicles, recommends that purified exosome fractions be simply designated as small EVs (sEV).

Exosomes have been subjected to intense research because their cargo is selectively internalized, and includes specific proteins and miRNA, which they deliver to adjacent and more distant cells [3,4]. The exosome cargo reflects the type and state of the cells, from which they originate, as do the molecules expressed in its limiting membrane. Some of the surface molecules (such as CD9, CD81 and CD63) are accepted as characteristic, if not completely exclusive, exosome markers that are expressed on most exosomes, regardless of origin. Other proteins, not specific for exosomes, are restricted to sEV derived from particular cell lineages, such as the asialoglycoprotein receptor on hepatocyte-derived exosomes [5], and CD31 and CD146 on those released by endothelial cells [6]. Finally, some molecules, such as Major Histocompatibility Complex Class I and Class II molecules, are expressed on EV membranes more generally [4,7,8]. It follows that characterizing exosomes in human blood, and other biological fluids, could provide insights into the nature and activity of systemic disease. However clinical application of this approach requires simple and robust methods for isolating exosomes from human blood, which is challenging [1,8,9].

Numerous techniques have been used to isolate exosomes from tissue culture supernatants [10,11,12,13,14,15,16,17,18,19,20], but purifying them from human blood is more difficult for at least two reasons: the EV in blood are far more heterogeneous than those generated in vitro, and second, they are suspended in plasma, which is more complex than tissue culture medium. Differential ultracentrifugation (DUC) is currently the standard method for isolating exosomes relatively free from contamination from other EV [20,21,22,23], but is cumbersome and requires relatively large sample volumes. The search for simpler and quicker methods led to strategies using size exclusion chromatography, such as commercially available Exo-spin™ columns (Cell Guidance Systems Ltd., Cambridge, UK) [17,24]. These isolate exosomes from culture supernatants are efficient and are simple enough for use in the clinic. It is now important to determine how efficiently and with what purity Exo-spin™ (EX05) columns purify exosome-containing sEV from human blood, and to compare results to isolation by DUC, which is what we have done.

We purified serum sEV from normal human donors by DUC and Exo-spin™ columns and compared the resulting exosome-rich fractions. Immunoblotting, ELISA and MACSPlex bead assays for CD9, CD63 and CD81 showed that both methods efficiently isolate sEV, and this was corroborated by particle counting by nanoparticle tracking analysis (NTA), which was 500-fold greater in the Exo-spin™ samples. Immuno-electron microscopy (EM) confirmed that the majority of sEV in the DUC were exosomes; by contrast, most particles in the Exo-spin™ samples co-purified lipoproteins and lipid droplets. Thus, Exo-spin™ columns effectively purify sEV from human serum, provided the influence of the attendant lipid contamination on the analytical strategies to be applied to the purified sEV is carefully considered.

## 2. Results

### 2.1. Nanoparticle Tracking Analysis of Serum-Derived sEV Purified by DUC or Exo-spin™

We quantified particles in the sEV preparations obtained by DUC or Exo-spin™ columns from nine fresh serum samples taken from six healthy donors. There were numerous particles in samples purified by both methods from all the sera (Appendix A). However, despite extraction from a 20-fold larger serum volume (2 mL versus 0.1 mL), there were far fewer particles in DUC preparations than in the Exo-spin™ ones: DUC—2.72 ± 2.22 × 10^8^ (Range: 0.51–8.07 × 10^8^) vs Exo-spin™—49.76 × 10^8^ ± 76.95 (Range: 0.51–254.67 × 10^8^); *p* = 0.03147, Wilcoxon rank sum test (Figure 1a). After correcting for the different serum volumes used, there was a mean of 500-fold more particles isolated per mL of serum by Exo-spin™ than by DUC: 497.56 ± 769.5 × 10^8^/mL of serum and 0.99 ± 1.11 × 10^8^/mL of serum, respectively (*p* = 4.114 × 10^−5^, Wilcoxon rank sum test) (Figure 1b). Similarly, the size distribution was narrower (Figure 1e–g), and the particle mean diameter smaller (Figure 1c) in the Exo-spin™ preparations: Exo-spin^TM^—132.52 ± 12.05 nm; and DUC—148.11 ± 14.64 nm (*p* = 0.0215, Wilcoxon rank sum test). Typically, the diameter of exosomes is less than 150 nm and the proportion of particles of this size was significantly lower in the DUC preparations (Figure 1d): DUC—57.66 ± 8.8%; Exo-spin™—69.61 ± 8.43% (*p* = 0.03147, Wilcoxon rank sum test). Consequently, the yield of small particles per ml of serum was significantly greater in Exo-spin™ purified samples than in DUC: Exo-spin™—404.41 ± 690.94 × 10^8^; DUC—0.57 ± 0.64 × 10^8^ (*p* = 4.114 × 10^−5^, Wilcoxon rank sum test). This suggests that Exo-spin™ purification is much more efficient than serum, but this needed to be verified by comparing the abundance of the exosome-specific proteins in the preparations, such as CD9, CD63 and CD81.

### 2.2. Analysis of the Abundance of Exosomal Markers by Western Blotting and ELISA

Nanoparticle tracking analysis counts all the particles in a sample and not just EV, and we used the abundance of three characteristic exosome proteins, CD9, CD63 and CD81 as surrogates to estimate yields on sEV isolated from serum. In striking contrast to the NTA, immunoblotting showed that concentrations of CD9 and CD63 were appreciable higher in the DUC-purified sEV samples (Figure 2a,b and Appendix A), suggesting that they contained more exosomes. Re-analysis of the samples by ELISA specific for CD9, CD63 and CD81 confirmed the result and showed the DUC preparations contained significantly higher mean concentrations of all three proteins (expressed in ELISA intensity units): CD9—0.96 ± 0.48 and 0.18 ± 0.07 for DUC and Exo-spin™ respectively (*p* = 0.031, Wilcoxon signed rank test); CD63—0.52 ± 0.25 and 0.16 ± 0.07 (*p* = 0.031, Wilcoxon signed rank test); and CD81—0.36 ± 0.18 and 0.12 ± 0.08 (*p* = 0.031, Wilcoxon signed rank test). Results from an ELISA for calnexin excluded significant contamination of the samples with intracellular contents (data not shown). Concentrations of CD9, CD63 and CD81 varied from donor-to-donor, but were invariably higher in the DUC and correlated closely with each other, regardless of the purification method (Figure 2b–d). Consequently, ranking the donors based on the concentrations of CD9 gave near identical results to ranking them according to CD63 or CD81 (Figure 2b–d). The greater abundance of exosome specific proteins in DUC samples, together with the results from individual donors provide strong evidence that CD9, CD63 and CD81 concentrations in a given sample all reflect the number of exosomes that it contains.

Next, we explored the obvious contradictions between NTA and protein abundance data by correlating the particle counts for individual donors with the concentration of CD9, CD63 and CD81 in the same sample. In the DUC preparations, the particle number correlated closely with the concentrations of all three exosome proteins: CD9—R² = 0.770 (*p* = 0.021); CD63—R^2^ = 0.548 (*p* = 0.092); CD81—R^2^ = 0.256 (*p* = 0.305); whereas there was no correlation in the Exo-spin^TM^ samples: CD9—R^2^ = 0.011 (*p* = 0.842); CD63—R^2^ = 0.058 (*p* = 0.644); CD81—R^2^ = 0.012 (*p* = 0.5) (Figure 2e–g and Appendix A). These data strongly suggest that the Exo-spin^TM^ purified samples contain large numbers of small particles that are not sEV and so we used electron microscopy to investigate their nature.

### 2.3. Ultrastructure of sEV Preparations Isolated by Exo-spin™ and DUC and Analysed by Transmission Electron Microscopy (TEM)

Samples from three donors were used to examine the ultrastructural appearances of sEV preparations purified by DUC and Exo-spin™. Many sEVs in the DUC samples had the distinctive cup shape previously reported to be characteristic of exosomes, but they were heterogenous in size and were often found in small groups with visible aggregates (Figure 3a,b). Immunostaining with gold-labelled specific antibodies showed that around 30% and 70% of the sEV expressed CD9 and CD63, respectively, confirming that they were exosomes (Figure 3c,e). The Exo-spin™ purified samples contained many more vesicles, which occurred in densely layered sheets and had a very different morphology to the DUC preparations. There were at least three distinct populations of particles (Figure 3d,f): dark vesicles with a high contrast membrane; smaller and brighter shade-like vesicles; and large vesicles with a single dark spot at one of the poles. Only rare vesicles expressed CD9 or CD63 in the immunogold stained preparations. These data confirm the inferences drawn from the protein expression data: in the DUC preparations, most of the particles identified by NTA are sEV. By contrast, the vast majority of vesicle-like particles purified by Exo-spin™ are not sEV, and so we determined whether they could result from co-purified serum proteins or lipids.

### 2.4. Co-Purified Lipoproteins Account for the Excess Particles in Exo-spin™ Preparations

The mean total protein concentrations in DUC and Exo-spin^TM^ preparations from five donors were 32.4 µg/mL ± 7.79 and 40.6 µg/mL ± 16.81, respectively. Silver stained SDS-PAGE of DUC samples showed a dominant band with the molecular mass, indicative of albumin, which equates to a concentration of less than 4.16 µg/mL if the designation is correct (Appendix A). We did not detect albumin in the Exo-spin^TM^ samples, but instead, the silver stained SDS-PAGE showed intense bands with masses >250 kDa, which is suggestive of lipoproteins; there were no corresponding proteins identified in the DUC preparations (Appendix A). Next, we measured concentrations of Apolipoprotein B (ApoB) by specific ELISA as a surrogate for lipoproteins more generally. The concentration of ApoB was similar in DUC and Exo-spin^TM^ purified sEV samples, with 0.39 ± 0.17 and 0.4 ± 0.23 ELISA units, respectively. However, the ApoB concentration correlated closely with the NTA particle count in the Exo-spin^TM^ samples (R² = 0.8538; *p* = 0.008), but not in those purified by DUC (R² = 0.3561; *p* = 0.211) (Figure 4a,b). Importantly, the serum triglyceride concentration correlated with the ApoB concentrations in the Exo-spin™ sEV preparations (R² = 0.71; *p* = 0.035), and with the NTA particle count (R² = 0.945; *p* = 0.001); there were similar, but less strong correlations between particle count and total serum cholesterol, HDL, LDL (Appendix A). None of these correlated in the DUC preparations (R² = 0.003; *p* = 0.91 and R² = 0.135; *p* = 0.473, respectively; Figure 4c–f).

These data strongly suggest that the excess particles in the Exo-spin™ samples result from co-purified triglyceride containing serum lipids, and this is supported by the EM images, since many of the particles are reminiscent of those reported for lipoproteins. In particular, those with the dark spot at one pole (Figure 5b,d) were strikingly similar to lipid droplets, in which triglycerides accumulate in a characteristic pocket-like structure, which has been described previously [25]. Immuno-EM using antibodies to ApoB revealed abundant gold particles in the Exo-spin™ preparations, whereas none of the vesicles in DUC preparations were ApoB positive. Unfortunately, the vesicles were too small and too abundant in the Exo-spin™ preparations to be certain, whether they were directly linked to the densely distributed gold particles (Figure 5). Nevertheless, we conclude that the excess particles in Exo-spin™ samples are serum lipoprotein complexes co-purified with the sEV, and it was important to know whether they influenced characterization of proteins expressed in the sEV limiting membrane.

### 2.5. MACSPlex Analysis of Membrane Proteins on sEV Purified by DUC and Exo-spin™

We used bead-based multiplex EV analysis by flow cytometry (MACSPlex Exosome Kit, human, Miltenyi Biotec) to characterize proteins expressed on the surface of the serum-derived sEV. The kit consists of a set of 39 hard-dyed capture bead populations, each of them coated with different monoclonal antibodies specific for one of 37 proteins reported to be expressed on the exosome surface, together with antibodies of two control proteins. The antibody-coated beads capture sEV expressing the relevant molecule, which can then be quantified by flow cytometry after tagging with a cocktail of labelled antibodies to CD9, CD63 and CD81. Paired samples of DUC and Exo-spin™ purified sEV from six donors were analyzed, and as expected, median allophycocyanin (APC) fluorescence intensities from the DUC samples were consistently higher (Figure 6a shows means (*n* = 6) of median APC fluorescence for each marker and individual median APC fluorescence intensities are shown in the Appendix A). In the DUC preparations, we obtained median APC fluorescence intensities above the limit of detection for 21 of the 37-exosome proteins, and 10 in the Exo-spin™ samples (Table 1). Next, we calculated the mean from all median APC fluorescence intensities for the proteins as an estimate of the abundance of exosomes in the preparation, and these were again significantly higher in the DUC preparations (Figure 6a): DUC—7124.80 ± 12,203.11; Exo-spin™—1462.42 ± 2451.13 (*p* = 0.00012, Wilcoxon signed rank test). These summary data are mirrored by the median APC fluorescence intensities of specific exosomal markers CD9, CD63 and CD83, which were also uniformly higher in the DUC preparations (Figure 6b–i): CD9—29,015.8 ± 16,072.4 versus 5194.5 ± 4542.6 (*p* = 0.031, Wilcoxon signed rank test); CD63—28,660.0 ± 12,303.0 versus 11,307.8 ± 8743.4 (*p* = 0.031, Wilcoxon signed rank test); and CD81—5905.2 ± 3686.9 versus 3763.8 ± 3586.7 (*p* = 0.16, Wilcoxon signed rank test). However, 20-fold more serum was used to purify sEV by DUC than by for the Exo-spin™ samples, which were also re-suspended at twice the dilution. Correcting for these differences showed that Exo-spin™ was at least as efficient as DUC at isolating sEV from human sera, and probably more so.

The MACSPlex assays also allowed us to characterize the populations of sEV in DUC and Exo-spin™ samples by comparing the median APC fluorescence intensities of all 37 exosome surface proteins. The previously described heat map (Figure 6) shows striking similarities between the DUC and Exo-spin™ samples, which we quantified by both correlating the mean APC signals for all proteins (multiple R = 0.85, *p* = 4.03 × 10^−12^), and by correlating the paired sEV samples from each of the donors (Multiple R = 0.81 ± 0.10 (range: 0.63–0.94)). The correlated abundance of individual markers is shown in Figure 7. Taken together, the MACSPlex data demonstrate that DUC and Exo-spin™ purify identical populations of sEV from a given serum sample. They also provide the initial data about the cellular origins of the sEV, since there were recurrent signals from multiple donors suggesting leukocytes (CD24, CD40, CD45, and CD56), endothelium (CD31, CD62P, and CD105), and platelets (CD41b and CD42a) were consistent sources. Coincidentally, the data also show that the co-purified lipoproteins do not interfere with antibody-based strategies for characterizing proteins expressed on the sEV membrane.

## 3. Discussion

The molecules expressed in the limiting membranes of circulating exosomes, together with cargo within their lumens, bear the imprint of the cell and tissue in which they originated, and whether it is stressed or injured [5,26,27]. These are highly attractive properties for non-invasive probes for human disease diagnosis and monitoring, but exploitation of exosomes as biomarkers that have been hindered by uncertainties about how best to isolate them from human blood. Here we report a detailed comparison of their isolation from human serum by DUC and the much simpler method of Exo-spin™ columns (Table 1). Our results show that: (i) both techniques efficiently purify similar populations of sEV from human serum, but larger numbers can be obtained by DUC; (ii) Exo-spin™ columns co-purify copious lipoprotein particles together with the sEV, and this invalidates the use of NTA to measure sEV yield, which can only be inferred from the abundance of exosome specific proteins, such as CD9, CD63 and CD81; and (iii) despite obvious donor-to-donor variation in abundance, MACSPlex analysis of sEV surface proteins uniformly demonstrate strong signals indicating their leukocyte, platelet, and endothelial origins. Our results provide essential information about purification and quality control of sEV from human serum and raise questions about the choice of method for particular studies.

Differential ultracentrifugation is a well established method for purifying sEV that can be applied to relatively large volumes of serum, but it is too laborious to be used for routine clinical patient monitoring, though not for research studies. By contrast, Exo-spin™ is simple enough for routine clinical use, but can only be applied to small volumes of serum (100 µL for the Exo-spin™ used in this study) [28]. Our data confirm the effectiveness of Exo-spin™ columns for isolating sEV from human sera and show that, based on the abundance of CD9, CD63 and CD81, they are at least as efficient as DUC, and probably more so, when the results had corrections had been made for the different volumes of serum used (100 µL and 2 mL respectively). Nevertheless, the relatively small volumes of serum that can be applied to Exo-spin™ columns limits analytical studies that can be applied to the purified sEV. This was highlighted by the MACSPlex analysis, which identified 21 different proteins in DUC purified sEV samples, but only 10 in those purified by Exo-spin™; the remaining 11 were below the limit of detection in the assay. Nevertheless, the similarity of the abundance profiles for the proteins identified by MACSPlex confirms that both techniques isolate identical populations of sEV. The failure to identify low abundance proteins of Exo-spin™ samples could be addressed by using a larger column, such as the Exo-spin™ EXO4, with a maximum volume of but the advantage of DUC is that there is no limit to volumes of serum that can be sampled, and thus it is possible to isolate as many sEV as is required.

Co-purification of other blood components with the sEV is a challenge for all the methods, and contamination with lipoproteins, and in particular VLDL and small chylomicrons, has previously been reported [20,29,30,31]. However, the lower density of lipids separates most of them from sEV during DUC, but their overlapping size ranges with sEV and their abundance in serum (1 × 10^12^/^mL^) [32] proved to be a major issue in the Exo-spin™ separations. This was highlighted by NTA, which showed that Exo-spin™ samples contained around 500 times more particles per unit volume of blood than the DUC preparations, despite their significantly lower concentrations of exosome specific proteins. The first evidence that high particle counts were co-purified lipoproteins came from the close correlation between particle number and ApoB concentrations in Exo-spin™ samples, and with serum triglycerides. We suspect that VLDL is the main source of the lipid particles, since they are normally between 30 and 90 nm in diameter [33]. The similarities between the ultrastructural appearances of IDL [34], VLDL [35], HDL [36] and the particles in Exo-spin™ samples strongly support this interpretation, which is highlighted by the presence of lipid droplets with their characteristic triglyceride pockets [25]; immunoEM with antibodies specific for ApoB confirmed the excess of lipoproteins in the Exo-spin™ samples, even though the density of the particles and the number of gold beads prevented us from directly linking the two. By contrast, most vesicles in the DUC preparations had the typical appearance of exosomes, albeit sometimes in clumps or aggregates, and immunoEM studies showed that many expressed CD9 and CD63, which confirmed their nature; only a very small fraction of vesicles in the Exo-spin™ preparations expressed these exosome markers. Thus, DUC results in relatively pure population of sEV whose yield can be estimated by NTA or from the abundance of CD9, CD63 and CD81. By contrast, the yield of sEV in Exo-spin™ preparations can only be estimated by the abundance of exosome specific proteins.

There were marked donor-to-donor variations in the yields of sEV. Blood was always drawn in the morning, but donors were not fasting and we did not control for physical activity, which could affect the concentration of circulating sEV [37]; nevertheless, these are unlikely to completely account for the differences we observed. It is striking that donor related differences in yield were conserved across the various measurement, regardless of the purification method, and whether the sEV yield was estimated by CD9, CD63 and CD81 abundance or MACSPlex analysis, or in the case of DUC, by NTA; this strongly favors a biological rather than a technical explanation. The MACSPlex data also provides initial data about the cellular source of the sEV in the serum with multiple recurrent signals suggesting leukocytes (CD24, CD40, CD45, and CD56), endothelium (CD31, CD62P, and CD105), and platelets (CD41b and CD42a) although the latter has been released ex-vivo. Nevertheless, these data provide strong encouragement for studies of sEV as biomarkers of disease activity.

In conclusion, we have shown that sEV can be purified efficiently from human serum by differential ultracentrifugation (DUC) and size exclusion chromatography (Exo-spin™ EX05 mini-HD), but that NTA cannot be used to calculate yields with the latter. The main differences between the methods are summarized in Table 2. Our data provide the foundation for the studies in the use of sEV as “liquid biopsies” to study human disease.

## 4. Materials and Methods

### 4.1. Blood Samples

Whole blood from healthy volunteers was drawn into VACUETTE^®^ TUBE Z No Additive (Greiner Bio-One, Austria), and left to clot for 20–30 min at room temperature before being centrifuged 2500× *g* for 15 min at room temperature.

### 4.2. Isolation of sEV by Differential Ultracentrifugation (DUC)

Samples of 2 mL of serum were diluted in 31 mL of cold PBS in order reduce the viscosity and albumin concentration before being subjected to differential ultracentrifugation (Allegra X-15R, Beckman Coulter, Brea, CA, USA; Ultracentrifuge CP100NX, Hitachi, Tokyo, Japan). Large vesicles were removed by ultracentrifugation at 30,000× *g* for 2 h at 4 °C. The supernatant was collected and sEV were pelleted at 100,000× *g* for 2 h, 4 °C. The supernatant was discarded and the pellet was resuspended in 33 mL of cold PBS before re-pelleting with a second ultracentrifugation step (100,000× *g*; 2 h; 4 °C) to wash the sEV. After discarding the supernatant, the sEV-containing pellet was gently resuspended and transferred to an Eppendorf tube, in which the volume was adjusted to a total of 200 µL by adding cold PBS. Initial experiments established DUC could be used to harvest sEV from between 0.5 to 10 mL serum (optimal volume 1 to 5 mL) (Appendix A); we used a standard volume of 2 mL for all the experiments reported here. Additionally, we determined the repeatability of this isolation method described by the number of recovered particles, size and distribution (Appendix A).

### 4.3. Isolation of sEV by Exo-spin™ Mini-HD Column EX05

Exo-spin™ mini-HD column EX05 (Cell Guidance Systems, Cambridge, UK)—in the text generally referred to as Exo-spin™—were used to isolate sEV from 100 µL of serum and resuspended in 400 µL of PBS, according to the manufacturer’s instructions. Briefly, 100 µL of sera were centrifuged for 30 min, 14,000× *g* and loaded onto the columns after equilibrating them with 3 × 2.5 mL PBS. After washing with 900 µL of PBS, the sEV fraction was eluted into an Eppendorf tube with 400 µL of PBS.

### 4.4. Nanoparticle Tracking Analysis

Nanoparticle tracking analysis captures videos of microscopic images of light scattering induced by Brownian motion of vesicles and protein aggregates. The videos are analyzed by the NTA software, which calculates the size of single particles and their concentration [32,38,39]. We used a Zetaview^®^ Basic NTA PMX 120 (Particle Metrix, Inning am Ammersee, Germany) system. All samples were freshly diluted in PBS, which was set as a background noise within these experiments. We evaluated the standard settings that were kept throughout the whole experiment: sensitivity = 68; shutter = 65; temperature = 24 °C. The dilution factor (DF) was changed for samples depending on an estimated particle concentration in samples (for DUC, DF = 25–150; for Exo-spin™, DF = 150–1500) in order to fit the concentration range: 10^5^–10^9^ particles/mL. For each measurement, three cycles were performed by scanning 11 cell positions each and capturing 30 frames per position. The images were analyzed with FlowJo_v10.6.1 (FlowJo LLC, Ashland, OR, USA) and R version 3.6.3 (R Core Team) for Windows.

### 4.5. Primary and Secondary Antibodies

The primary antibodies used in transmission electron microscopy and Western blotting and ELISA were as follows: mouse anti-CD63 (10628D; Invitrogen, Waltham, MA, USA); mouse-anti CD9 (AHS0902, Invitrogen, Waltham, MA, USA); and mouse anti-ApoB100/48 (3715-3-250, MabTech, Nacka Strand, Sweden). Antibodies anti-CD81 (10630D, Invitrogen, Waltham, MA, USA) and anti-Calnexin (C5C9) (#2679, Cell Signaling Technology, Danvers, MA, USA) were only used on ELISA.

The secondary antibody used in transmission electron microscopy (TEM) was gold-labelled (EM:GAM15, goat anti-mouse 15 nm immunogold-conjugate; BBI Solution, Crumlin, UK). For Western blotting, we used fluorescently labelled secondary antibodies: goat anti mouse IRDye^®^ 800 or goat anti mouse IRDye^®^ 680 or goat anti rabbit IRDye^®^ 680 or goat anti rabbit IRDye^®^ 800 (LI-COR Biosciences, Lincoln, NE, USA). The ELISA secondary antibodies were conjugated with horseradish peroxidase (HRP) as follows: Goat anti rabbit- HRP (111-036-047, Jackson Immunoresearch, West Grove, PA, USA); or Rabbit anti mouse-HRP (JZM035046_Fa, Ancell Immunology Research, Bayport, MN, USA).

### 4.6. Transmission Electron Microscopy and Immuno-Electron Microscopy

Transmission electron microscopy was performed, as described by Thery et al. [16]. Briefly, the small extracellular vesicle (sEV) preparations were fixed in 2% PFA and a 5 µL drop was placed on formvar-coated 200 mesh Ni-grids (S162N; Agar Scientific Ltd., Standsted, UK). CD63 and CD9 was detected in sEV by immuno–electron microscopy after incubating the grids on drops of the antibody dilution in 1% Eggalbumin overnight in a humid chamber at 4 °C with mouse anti-CD63 at a 1:40 dilution, mouse- anti CD9 at a 1:20 dilution, or mouse anti-ApoB100/48 at a 1:10 dilution. Binding of antibodies was detected with a gold-labelled secondary antibody goat anti mouse-diluted 1:50. Washing steps between antibody incubation were performed with PBS followed by post-fixation with 1% glutaraldehyde, incubation in a solution of neutral uranyl oxalate and covering with a polyvinylalcohol-film. After air-drying, the grids were examined in a TEM JEOL 1400 PLUS (JEOL, Tokyo, Japan) at 60 kV. Pictures were taken in Radius-software with Quemesa_Camera (Fa. Olympus, Tokyo, Japan).

### 4.7. SDS-PAGE and Western Blotting

Small extracellular vesicle preparations were purified by differential ultracentrifugation and Exo-spin™. 30 µL of each sEV sample was mixed with 10 µL 4 × Protein Loading Buffer (LI-COR, Biosciences, Lincoln, NE, USA) and incubated at 95 °C for 5 min. The samples were separated on either 10% or 4–20% gradient SDS-PAGE non-reducing conditions. In some conditions, dilutions of human serum, lysates of THP-1 cells or purified albumin were added as protein loading controls. Cells were lysed in RIPA buffer (150 mM NaCl; 5mM EDTA, pH 8.0; 50 mM TRIS, pH 8.0; 1% Triton X-100; 0.5% sodium deoxycholate; 0.1% SDS) by sonication.

Silver staining of SDS gels was performed with PlusOne™ Silver staining kit, Protein (GE Healthcare Bio-Sciences AB, Stockholm, Sweden) according to the manufacturer’s instructions. For Western blot analysis, gels were transferred onto a nitrocellulose membrane (GE Healthcare, Chicago, IL, USA). The membranes were blocked for 1h in RT with Intercept^®^ (TBS) Blocking Buffer (LI-COR, Biosciences, Lincoln, NE, USA) or Odyssey^®^ (PBS) Blocking Buffer (LI-COR, Biosciences, Lincoln, NE, USA). Incubation with primary antibodies to either CD63 or CD9 was performed under constant shaking for 1 h at RT or overnight at 4 °C. Next, the membranes were washed with TBST (TBS + 0.1% Tween20) and incubation with the appropriate secondary antibody was performed for 1 h at RT (goat anti mouse or goat anti rabbit). Three washing steps between and after each antibody incubation were followed by visualization using an Odyssey Infrared Imager (LI-COR, Biosciences, Lincoln, NE, USA). Analysis was performed with ImageJ Studio Lite Version 5.2. (Rasband, W.S., ImageJ, U.S. National Institutes of Health, Bethesda, MD, USA)

### 4.8. ELISA

ELISA was performed using polystyrene 96-well plates (Nunc™, Roskilde, Denmark) with the following antibodies: anti-CD63; anti-CD9; anti-CD81; anti-Calnexin (C5C9); anti-ApoB100/48. Purified sEV were mixed with coating buffer (0.05 M Carbonate buffer; pH 9,6) at a ratio of 1:4 and incubated in the plates for 2 h at RT or overnight at 4 °C on a shaker. After washing the wells 3 times with TBST (TBS, 0.05% Tween20) for 5 min, blocking was performed with TBS containing 1% BSA for 2 h at RT. Primary antibodies diluted in blocking buffer were applied and incubated for 2 h at RT or overnight at 4 °C on a shaker. After 3 washes the appropriate secondary antibodies conjugated to HRP were applied to the plates followed by 2 h incubation at RT on a shaker. Antibody binding was visualized by obtaining 2,3-diaminophenazine and color reaction was stopped with 8N H_2_SO_4_. The absorbance was measured at 490 nm and 630 nm on a plate reader. The blanks were filled with only development buffer and stop buffer. As positive controls we used dilution of human serum and THP-1 cell lysates. As negative controls on the coated plated we applied only secondary antibodies (omitted primary antibody incubation) and only primary antibodies on a coated plate (omitted secondary antibody incubation). The intensity was calculated by subtracting 490 blank by 630 blank (OD), and we evaluated the mean intensity from 3 replicates for each sample—in the figures referred to as “Mean Intensity (OD)”.

### 4.9. MACPlex Human Exosome Pan Kit

The analysis was performed using the MACSPlex Exosome Kit, human (Miltenyi Biotec, Bergisch-Gladbach, Germany) according to the manufacturer’s instructions. Briefly, small extracellular vesicle preparations (120 µL) were incubated overnight with MACSPlex Exosome Capture Beads (15 µL; containing 39 different antibody-coated bead subsets) on an orbital shaker (450 rpm) at RT without access to light. Next, samples were washed by adding 500 µL of MACSPlex buffer into each tube and centrifuged at RT for 5 min 3000× *g*. The supernatant was discarded, and the pelleted beads were incubated with MACSPlex Exosome Detection Reagent (APC-conjugated anti-CD63, anti-CD9, anti-CD81) for 1 h at RT on a shaker (450 rpm). After this, beads were washed twice with MACSPlex buffer and fluorescence intensities for FITC, PE and APC were acquired by the BD FACSCanto II Cell Analyzer (BD Biosciences, Franklin Lakes, NJ, USA), resulting in approximately 5000–20,000 single bead events being recorded per sample. Analysis was performed in FlowJo_v10.7.1. by separating the bead populations according to their fluorescence properties in the PE and FITC channel. Median APC fluorescence was used to quantify the exosomes bound to the beads.

### 4.10. Micro BCA™ Protein Assay Kit

We determined the total protein concentration in the small extracellular vesicle preparations; we used the Micro BCA™ Protein Assay Kit (23,235, Thermofisher Scientific, Waltham, MA, USA) following the user guide for the microplate procedure. The selected samples were measured in duplicate and in one or two dilutions, depending on the volumes available.

### 4.11. Statistical Analysis

Statistical analysis was performed with R version 3.6.3 for Windows (R Core Team (2020). R: A language and environment for statistical computing. R Foundation for Statistical Computing, Vienna, Austria. URL https://www.R-project.org/and Microsoft Excel (Redmond, WA, USA). We applied Wilcoxon signed rank test and Wilcoxon rank sum test. The graphs were performed in Graphpad Prism 9.0.0 (GraphPad Software, San Diego, CA, USA).

## Figures and Tables

**Figure 1 ijms-22-04653-f001:**
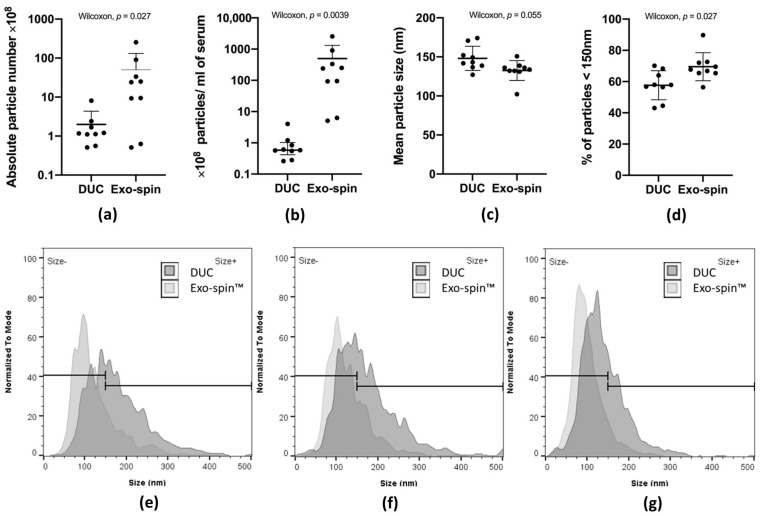
The mean particle number in sEV preparations isolated from nine serum samples withdrawn from 6 individuals. The graph (**a**) presents the total particle number in sEV suspensions isolated from different volumes of serum, respectively: Exo-spin™ 100 µl and DUC 2000 µl; the mean particle number in sEV samples normalized to the volume of serum used for isolation is shown on graph (**b**); image (**c**) shows the mean size of vesicles in nanometers for DUC and Exo-spin™ preparations. Percentage of vesicles smaller than 150 nm is displayed on graph (**d**); representative size distribution of analyzed sEV preparations isolated by DUC and Exo-spin™ (**e**–**g**). Statistical analysis was performed in R with Wilcoxon signed rank test for paired samples. The middle bar of each group shows the median with the error bars corresponding to the interquartile range. Abbreviations: DUC: differential ultracentrifugation.

**Figure 2 ijms-22-04653-f002:**
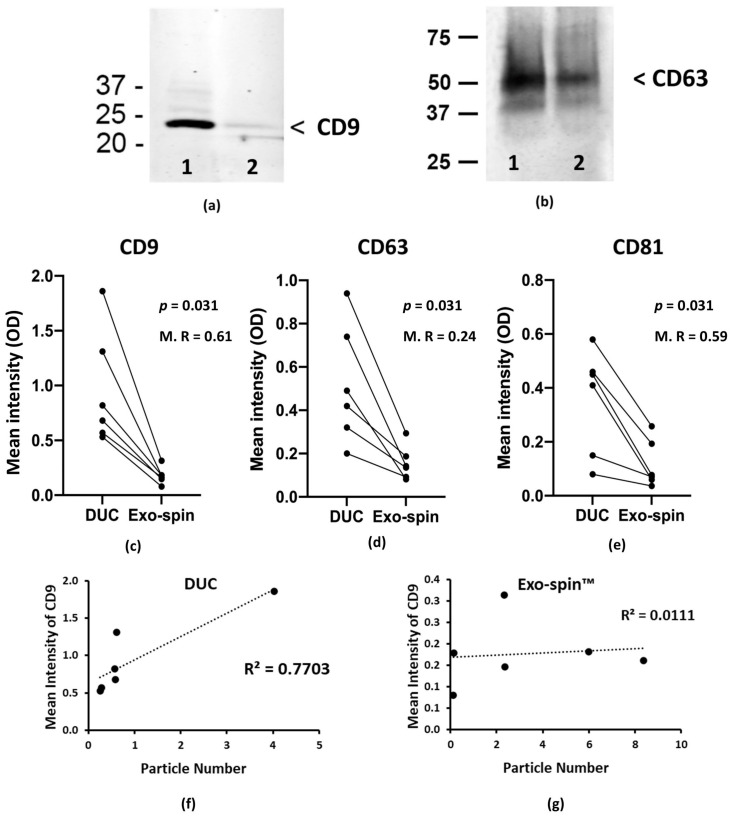
Image (**a**) shows a representative Western blot with antibody to CD9, and on a duplicate sample (image (**b**), to CD63 on small extracellular vesicles purified from one donor by either differential ultracentrifugation (lane 1) or Exo-spin™ (lane 2). ELISA results given in relative mean intensity (OD of measured absorbance (490 nm–630 nm)) are shown on the axis y of graphs (**c**–**e**) for exosome specific markers (CD9, CD63, and CD81). Graphs present the mean intensity from 3 replicates of the small extracellular vesicle (sEV) preparations purified from six individuals by both methods—individual results of paired samples are connected by a line. Graphs (**f**,**g**) show the correlation between the CD9 mean intensity and particle numbers calculated by nanoparticle tracking analysis for DUC and Exo-spin™ samples, respectively. Statistical analysis was performed in R with a Wilcoxon signed rank test. Abbreviations: M. R = Multiply R.

**Figure 3 ijms-22-04653-f003:**
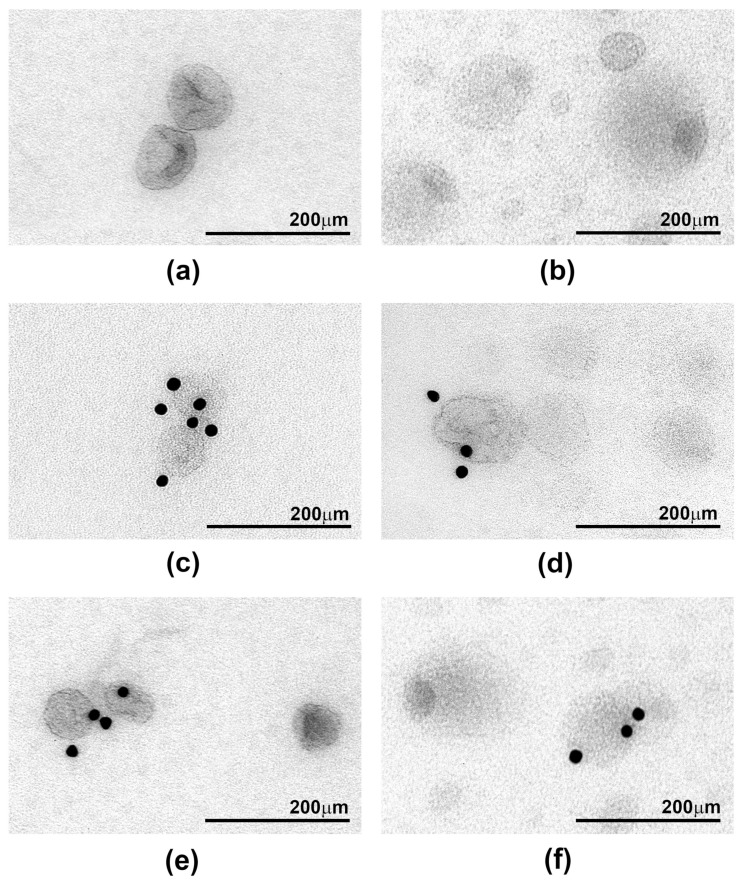
Transmission electron microscopy images of small extracellular vesicle (sEV) preparations obtained by two methods: differential ultracentrifugation (**a**,**c**,**e**) and Exo-spin™ (**b**,**d**,**f**). sEV preparations were identified using antibodies specific to either CD9 (**c**,**d**) or CD63 (**e**,**f**) and antibody binding was confirmed by secondary antibodies conjugated to 15 nm gold particles.

**Figure 4 ijms-22-04653-f004:**
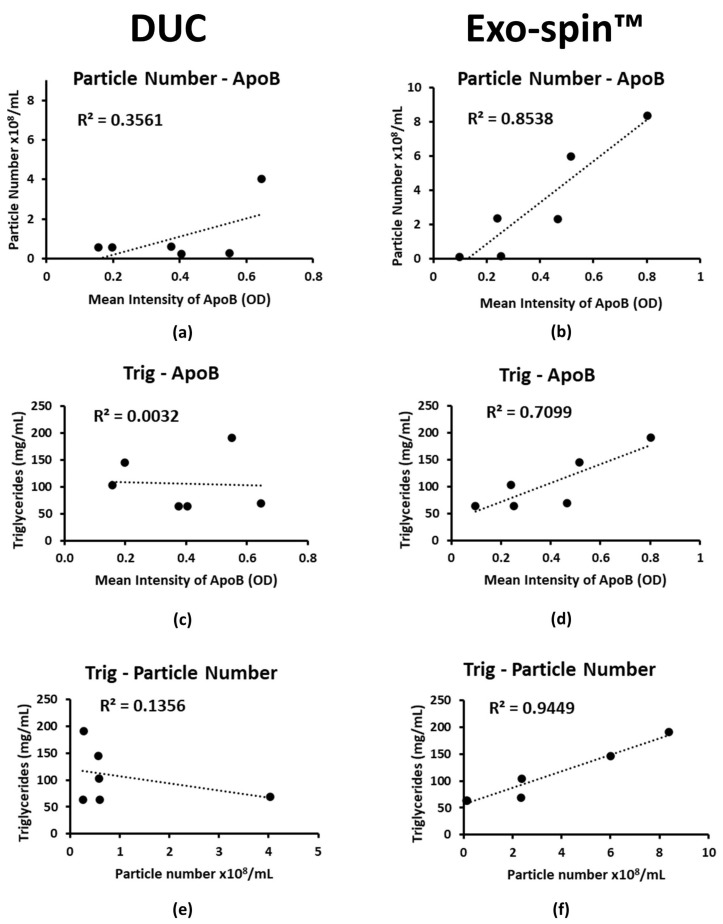
Lipoprotein contamination in small extracellular vesicles (sEV) preparations purified by differential ultracentrifugation (left column) and Exo-spin™ (right column). Graphs (**a**,**b**) present the correlation between isolated particle numbers (y axis) and mean Apolipoprotein B (ApoB) intensity (x axis) measured by ELISA in sEV preparations. Graphs (**c**,**d**) show the correlation between triglyceride concentrations (y axis) measured in serum prior to isolation and ApoB intensity (x axis) in sEV preparations. Graphs (**e**,**f**) show the correlation between triglyceride concentrations (y axis) measured in serum prior to isolation and particle number in sEV preparations (x axis). Statistical analysis of regression was performed in Microsoft Excel with the Data Analysis Tool.

**Figure 5 ijms-22-04653-f005:**
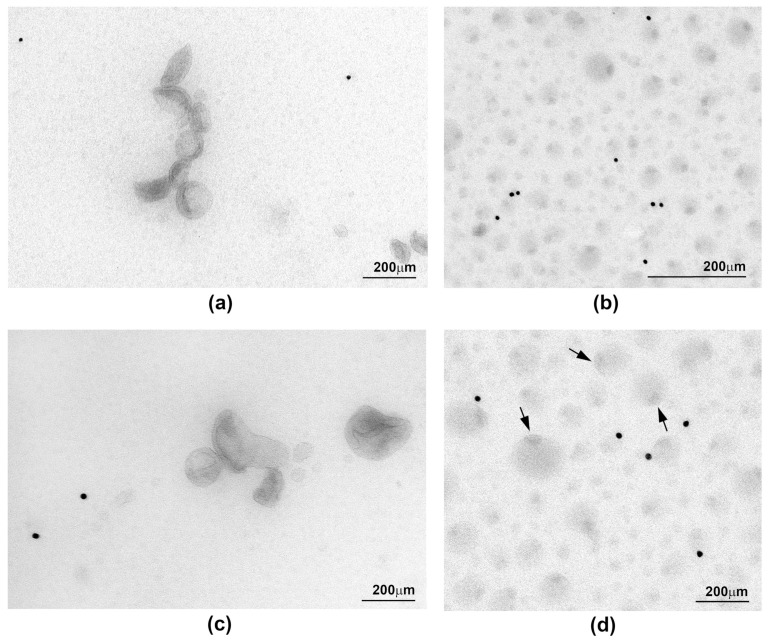
Transmission electron microscopy images of small extracellular vesicle preparations purified by differential ultracentrifugation (DUC) (**a**,**c**) or Exo-spin™ (**b**,**d**) that were incubated with an antibody specific for Apolipoprotein B (ApoB). Antibody binding was detected using a secondary antibody conjugated to 15 nm gold particles. The arrows indicate vesicles that resemble lipid droplets with a triglyceride pocket.

**Figure 6 ijms-22-04653-f006:**
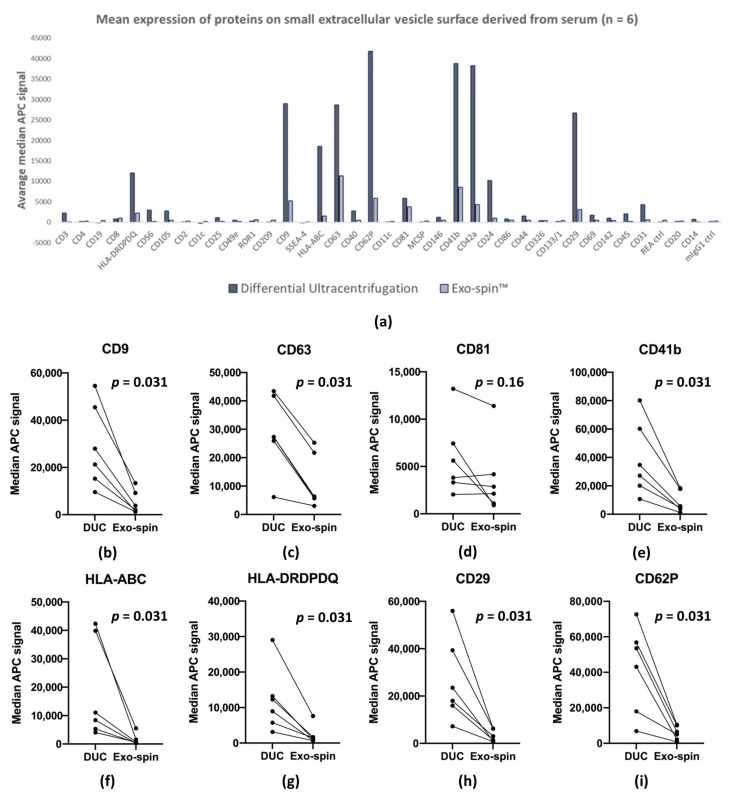
MACSPlex assay of small extracellular vesicle (sEV) preparations purified by Exo-spin™ and differential ultracentrifugation (DUC). (**a**) presents mean of median APC fluorescence intensities shown on axis y (*n* = 6) for all markers included in the kit (x axis). (**b**–**i**) show median APC fluorescence intensities (y axis) from sEV preparations purified from 6 donors (each dot = 1 donor) and paired with the corresponding samples obtained by the other method (x axis). Statistical analysis was performed in R with Wilcoxon signed rank test for paired samples.

**Figure 7 ijms-22-04653-f007:**
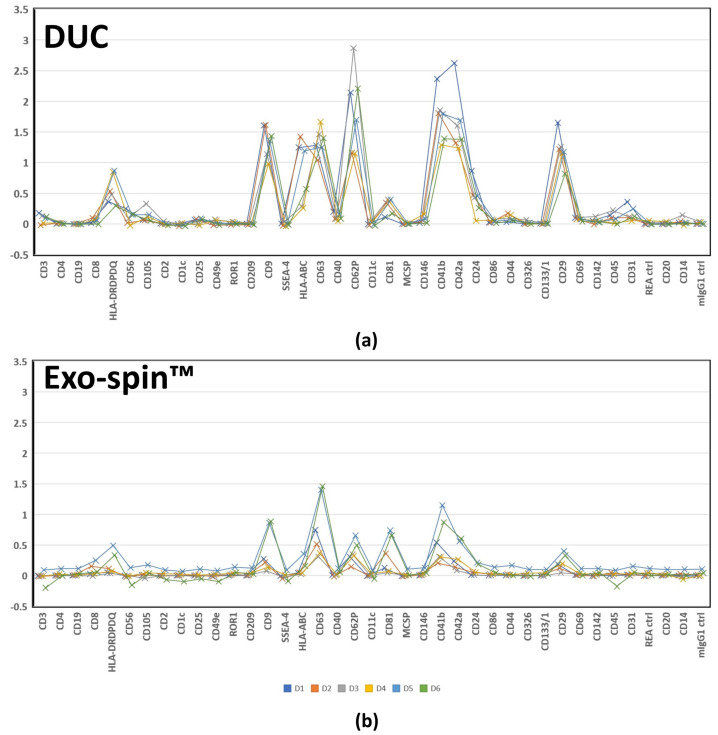
Correlation between 39 proteins median allophycocyanin (APC) fluorescence intensities acquired by bead-based multiplex EV analysis (MACSPlex Exosome Kit, human, Miltenyi Biotec) of small extracellular vesicle sEV) preparations purified by differential ultracentrifugation (**a**) (DUC) and Exo-spin™ (**b**). The median APC fluorescence intensities of 39 proteins shown on axis y were normalized to the mean of CD9, CD63 and CD81 median APC fluorescence intensity of each sEV sample. Each colored line corresponds to one donor and the samples between methods are paired. The specific markers are shown on axis x.

**Table 1 ijms-22-04653-t001:** Mean median allophycocyanin (APC) fluorescence intensities of markers detected by MACSPlex kit in preparations of small extracellular vesicles purified by differential ultracentrifugation (DUC) and Exo-spin™ (paired samples from 6 donors). The negative controls are marked in grey color. The markers were regarded as present when their median APC fluorescence intensities were above 1000 procedure defined units.

Marker	DUC	Exo-spin™
CD62P	41,826	5839
CD41b	38,839	8594
CD42a	38,260	4270
CD9	29,016	5195
CD63	28,660	11,308
CD29	26,669	3089
HLA-ABC	18,512	1518
HLA-DRDPDQ	12,069	2206
CD24	10,192	957
CD81	5905	3764
CD31	4360	606
CD56	2969	153
CD105	2781	512
CD40	2759	451
CD3	2291	33
CD45	2090	191
CD69	1721	482
CD44	1522	525
CD146	1234	502
CD25	1121	221
CD142	1011	417
CD86	788	481
CD8	765	1052
CD14	679	75
CD49e	466	155
CD326	369	360
ROR1	291	556
mIgG1 ctrl	210	281
CD133/1	156	378
CD4	139	320
CD20	132	323
REA ctrl	128	504
CD2	85	265
MCSP	55	258
CD209	36	451
CD11c	0	233
CD19	−16	352
SSEA-4	−22	4
CD1c	−198	161

**Table 2 ijms-22-04653-t002:** Summary of small extracellular vesicles properties purified by differential ultracentrifugation (DUC) and Exo-spin™.

Characteristics	DUC	Exo-spin™	Comments
Minimal serum volume for healthy individuals	1 mL	100 µL	DUC range: 1–5 mL
			Exo-spin™ range: x–150 µL
Final volume of sEV resuspended in PBS	200 µL	400 µL	Volume of DUC can be adjusted for individual purposes
Total protein concentration	32.4 ± 7.8 µg/mL	40.6 ± 16.8 µg/mL	
Recovery (particle number/mL of serum)	1.0 ± 1.1 × 10^8^/mL	497.6 ± 769.5 × 10^8^/mL	High recovery of particles by Exo-spin is mostly associated with large numbers of lipoproteins being co-isolated;
			Normalisation is essential since the initial applied volume of serum differs
Absolute number of particles	2.7 ± 2.2 × 10^8^	49.8 ± 76.9 × 10^8^	Absolute number of particles that is purified from heathy individuals according to the suited protocol
Particle size (diameter) ø	Mean = 148.1 ± 14.6 nm	Mean = 132.5 ± 12.1 nm	Large size of particles isolated by DUC may be associated with aggreagates induction by gravitational forces
	Mode = 121.9 ± 20.6 nm	Mode = 110.9 ± 17.2 nm	
	Median = 138.0 ± 13.7 nm	Median = 123.1 ± 13.1 nm	
	% of particles < 150 nm = 57.66 ± 8.8%	% of particles < 150 nm = 69.6 ± 8.4%	
Relative mean intensity by ELISA	CD81: 0.36 ± 0.18;	CD81: 0.12 ± 0.08;	Measured particle concentrations for these samples were: DUC = 10.54 × 10^8^/mL; Exo-spin™ = 32.25 × 10^8^/mL
	CD63: 0.52 ± 0.25;	CD63: 0.16 ± 0.07;	
	CD9: 0.96 ± 0.48	CD9: 0.18 ± 0.07	
Median APC fluorescence intensity by MACSPlex	Average: 7124.8 ± 12,203.1	Average: 1462.4 ± 2451.1	All samples were paired
	CD81: 5905.2 ± 3686.9	CD81: 3763.8 ± 3586.7	
	CD63: 28,660 ± 12,303	CD63: 11,308 ± 8743.4	
	CD9: 29,016 ± 16,072	CD9: 5194.5 ± 4542.6	
Morphology	Singles and aggregates of vesicles, cup-like shape, approximately half of vesicle were stained positively with exosomal markers	Whole image fully layered with vesicles, various morphologies that does not resemble exosomes, only few stained positively with exosomal markers	-
Co-purification	Low lipoprotein contamination	High lipoprotein contamination	Most of the particles isolated by Exo-spin™ from non-fasted patients were assigned as lipoproteins;
	Albumin contamination approximately 12.85%	Non-detectable albumin contamination	Albumin contamination was evaluated by silver staining.

Abbreviations: APC—allophycocyanin; DUC—differential ultracentrifugation; ELISA—enzyme-linked immunoassay; sEV—small extracellular vesicles.

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
