# Peer review of "Isolation of Small Extracellular Vesicles from Human Sera"

_ijms, 2021, doi:10.3390/ijms22094653_

Round 1
Reviewer 1 Report
In the work of Małys et al, the authors compared the two different methods for the isolation of sEV from serum: differential ultracentrifugation (DUC) and exclusion chromatography using commercially available Exo-SpinTM columns. They reported that Exo-spinTM enables isolation of sEV from blood with vesicle populations similar to the ones recovered by DUC but with lower concentrations.
All the methods were well described by the authors and results were clearly exposed.
However, a table would better summarize the results obtained concerning the number of particles and the size of EVs in each sample obtained from different preparations.
The authors focused their analysis on three different exosomal markers: CD9, CD63 and CD81. Anyway, the concept to discriminate the different populations of EVs is very tricky and so that could be further validated, either by validating the reported markers also by mass-spectrometry analysis, or by enlarging the number of markers to be evaluated (i.e. annexins or tetraspanins).
Regarding western blotting, a better image should be presented regarding CD63; a bigger band could be obtained by using a higher amount of EVs for this test (30ul were used, but it would be preferable to choose the volume basing on the protein concentration or on the number of EVs).
Lastly, the NTA present some limitations, as the authors mentioned, together with the fact that this technique is not so sensible to sort EVs basing on the size. Another system should be used, for example by nano-flow cytometry (which discriminates different populations of EVs dependently on their size) or even MACSPlex analysis by flow cytometry that the authors employed for the evaluation of EV-markers.
Line 120: CD8 is CD81.
Reviewer 2 Report
The paper "Isolation of small extracellular vesicles from human sera" written by Malys et al compares two different methods for isolation of sEV from blood sera by differential ultracentrifugation and exclusion chromatography using commercially available Exo-SpinTM columns, with report that Exo-spinTM enables isolation of sEV from blood with vesicle populations similar to the ones recovered by DUC but with lower concentrations.
Concerns:
1. The authors should include the references because they didn't appear
2. In the graphs they must include the standard deviation
Round 2
Reviewer 1 Report
The authors have answered to the presented questions. They added supporting material required to the paper. From my point of view, the paper is now ready for publication.
Reviewer 2 Report
The paper was improved with the performed changes